# How Azanucleosides Affect Myeloid Cell Fate

**DOI:** 10.3390/cells11162589

**Published:** 2022-08-19

**Authors:** Anna Stein, Uwe Platzbecker, Michael Cross

**Affiliations:** Department of Hematology, Cell Therapy and Hemostaseology, Leipzig University Hospital, 04103 Leipzig, Germany

**Keywords:** hypomethylating agents, decitabine, azacytidine, epigenetics, hematopoiesis, myeloid neoplasia

## Abstract

The azanucleosides decitabine and azacytidine are used widely in the treatment of myeloid neoplasia and increasingly in the context of combination therapies. Although they were long regarded as being largely interchangeable in their function as hypomethylating agents, the azanucleosides actually have different mechanisms of action; decitabine interferes primarily with the methylation of DNA and azacytidine with that of RNA. Here, we examine the role of DNA methylation in the lineage commitment of stem cells during normal hematopoiesis and consider how mutations in epigenetic regulators such as *DNMT3A* and *TET2* can lead to clonal expansion and subsequent neoplastic progression. We also consider why the efficacy of azanucleoside treatment is not limited to neoplasias carrying mutations in epigenetic regulators. Finally, we summarise recent data describing a role for azacytidine-sensitive RNA methylation in lineage commitment and in the cellular response to stress. By summarising and interpreting evidence for azanucleoside involvement in a range of cellular processes, our review is intended to illustrate the need to consider multiple modes of action in the design and stratification of future combination therapies.

## 1. Introduction

The nucleoside analogues 5-aza-2′-deoxycytidine (decitabine, DEC) and 5-azacytidine (vidaza, AZA) are used widely in the therapy of myeloid neoplasia. Following incorporation into the DNA of cycling cells, the aza (N) substitution at position C5 of the pyrimidine ring results in the capture and cross-linking of DNA methyltransferase (DNMT) enzymes normally responsible for cytosine C5 methylation. At high drug doses, the resulting DNA enzyme adducts induce a cytotoxic DNA damage response with low specificity for cancer cells. This led to the azanucleosides being largely disregarded as anticancer agents following their original synthesis in the 1960s [1]. However, it later became clear that much lower doses of azanucleosides do have specific anti-leukemic effects and that these are associated with the reduction in cellular DNMT activity that is generally assumed to “correct” the aberrant DNA methylation of leukemic cells, and thus, restore them to a non-neoplastic fate. Based on this, the first successful trials of protracted, low dose DEC to treat myeloid leukemia were conducted in the early 2000s [2]. Since then, both DEC and AZA, now referred to collectively as “hypomethylating agents” (HMA), have been widely adopted in the therapy of myeloid neoplasia, where they have proven to provide benefit in a variety of situations in which aggressive chemotherapy is impractical and more targeted approaches are still lacking.

Although the benefit brought by HMAs can sometimes be long-term, responses are by no means universal, and relapse is common. For instance, of those elderly patients with high-risk myelodysplasia who are unable to undergo stem cell transplantation, only around half show an initial response to HMAs and the majority of these subsequently undergo relapse [3]. Efforts to improve the prospects for such patients are currently focussing on the use of drug combinations in which an HMA is given together with one or more agents that target specific cellular processes [4]. The results of some of these studies are very encouraging, but have highlighted a shortfall in our understanding of the routes through which the HMAs actually act to revert the leukemic phenotype. To date, the assumption has often been that both DEC and AZA work in more or less the same way. However, while they demonstrate broadly similar clinical efficacies as monotherapies for myeloid neoplasia, these are by no means identical [5], and the two agents clearly have distinct effects on target cells that relate to their respective DNA (DEC) and RNA (AZA) identities [6,7]. As HMA-based treatments become more common and more diverse, it is going to be increasingly important to understand precisely how the HMAs affect cell biology. This information will be key to interpreting the results of current and future trials and to determining which variables influence the probability, depth and duration of a clinical response in any given situation. Here, we aim to provide relevant background information by considering our current understanding of HMA function from a cell biology perspective.

## 2. DNA Methylation and the DNA Methyltransferases

DNA methylation is an evolutionarily ancient process, with DNMT homologues being present throughout the animal, plant and fungal kingdoms, traceable all the way back to prokaryotes [8]. DNMTs target cytosine in the context of the dinucleotide palindrome 5′-CpG-3′, resulting in methylation of the C residues on both strands of double-stranded DNA. Targeting a palindrome in this way ensures retention of the methylation mark on one strand of each duplex following replication, thus making CpG methylation marks heritable through cell division. Indeed, the enzyme DNMT1 is dedicated purely to restoring the second methyl group at such hemi-methylated sites, while DNMT3A and 3B are responsible for de novo methylation of previously non-methylated sequences.

In eukaryotes, methylation of CpGs is generally associated with a reduction in the expression of associated genes, initially because methylation blocks DNA binding by sequence-specific transcription factors, and subsequently because extensively methylated DNA tends to be packaged into inaccessible chromatin structures (Figure 1). The advantages of partitioning chromatin in this way most probably have to do with the challenges of managing large, complex genomes, particularly in multicellular organisms. The packaging of large parts of the genetic repertoire into structures that are inaccessible for the transcription machinery allows resources to be focused exclusively on the subset of genes that are relevant to a given situation or cell type and avoids interference by those that are irrelevant or potentially disruptive. The lineage commitment and differentiation of hematopoietic stem cells provides a good example of this. Here, multipotent stem and progenitor cells maintain genes relevant to their various multiple lineage fates in an accessible but repressed state of priming. Lineage commitment, then, occurs not just as the upregulation of those genes relevant to their chosen fate, but as the inactivation of those that are relevant to all other fates and could therefore interfere [9,10,11]. DNA methylation plays a major role in this reorganisation and expression of the de novo methylase DNMT3B is increased accordingly at this stage of hematopoiesis [12].

The special role of CpGs as targets of methylation makes them potentially disruptive when in the wrong place, partly because of the suppressive effect of methylation itself, and partly because 5-methylcytosine (5meC) can deaminate to thymidine, thus making the DNA prone to point mutation [13]. Accordingly, evolution has selected against CpGs in most of our genome, leaving isolated groups known as “CpG islands” [14] only where they have a useful role to play in influencing gene expression. Within a CpG island, transcription factors can be recruited to their non-methylated binding sites where they act in conjunction with a range of histone-modifying enzymes to establish an open and active chromatin structure (Figure 1). However, should the relevant transcription factors be depleted, the CpGs in their respective binding sites are left unprotected and susceptible to progressive methylation that then obstructs reactivation. Even so, methylation does not lead unavoidably to a downhill spiral of suppression, since a CpG island typically consists of multiple, interacting sites, and DNA methylation by the DNMTs can be counteracted by demethylation driven by enzymes of the ten-eleven translocation (TET) family of methylcytosine dioxygenases. It is, therefore, the dynamic balance between DNMT-driven methylation and TET-driven demethylation that mediates dynamic reorganisation one way or the other (Figure 1).

Since CpG methylation patterns are transmitted through rounds of DNA replication, they allow regulatory regions that control gene expression to be either maintained in an open, active state, or marked for closure in a way that is heritable through cell division. This provides a mechanism for the inheritance of phenotype that accompanies the inheritance of genotype from mother cell to daughter cell.

## 3. The Epigenetic Landscape

The idea that information can be inherited not just in the form of genes but also as levels of gene activity was first formalised by the developmental and theoretical biologist Conrad Waddington [15]. Despite having no knowledge of the genetic code, gene structure or regulatory mechanisms, Waddington proposed that units of genetic information (the genes themselves) are passed on together with an epigenetic (“above genetics”) pattern of chemical interactions between them. He represented the genes as an array of fixed coordinates at ground level and the interactions between them as a framework. Draped over and shaped by the framework is the now iconic landscape of hills and valleys mapping out the alternative routes available to a cell during development (Figure 2).

More than 60 years later, it is still helpful to use Waddington’s landscape as a map against which to summarise what we have learned in the meantime about development and regeneration at the cellular and molecular level. Considering the hematopoietic system as an example, the earliest multipotent stem cells would be situated at the top of the valley in a state of maximum potential that we now know to be reflected by widespread accessibility of chromatin. At this stage, genes relevant to all of the potential paths that can be taken are held in a primed state [9,10,11]. Waddington’s framework of chemical interactions that determines the topology of the landscape is actually a network of interacting transcription factors. These interactions guide sequential lineage decisions at each bifurcation by activating one programme and inactivating others, effectively throwing a succession of gene regulatory switches. The dynamic methylation/demethylation of transcriptional regulatory sequences is a central part of the switching mechanism that determines which networks are turned up and which turned off at each stage.

The differentiation of hematopoietic progenitors is accompanied by extensive proliferation and, as detailed above, the daughter cells are held on course through each division by epigenetic inheritance of the gene expression patterns from the mother cell. In principle, epigenetic memory could be passed on by any feature that can both contribute to cell-specific gene expression and survive a round of cell division. This includes some transcription factors [16], RNAs [17] and histone modifications [18]. For the most part, however, it is the DNA methylation state described in Figure 1 that carries epigenetic identity through cell division.

A key feature of the epigenetic landscape is that once a cell has left the stem cell pool, it’s journey is inexorably downhill. There is normally no point at which it can proliferate without progressing further through differentiation. To achieve this, the regulatory networks must be arranged to ensure that cell division is hard-wired to differentiation, allowing hematopoiesis to provide a constant flow of mature cells without blockages or back-ups. To achieve a leukemic state, this rule must be broken. One can envisage here how the framework of regulatory interactions can be distorted to create a “pocket” in the landscape in which it is possible for a cell to divide without further differentiating [19]. In theory, distortion of the framework in this way could be achieved at either the genetic or epigenetic level. However, since we know that leukemias have typically accumulated multiple genetic mutations before they become clinically apparent [20], it would appear that overt transformation requires a substantial shift in the genetic foundations, rather than a mere rewiring of the epigenetic connections between them. This is consistent with there being a strong evolutionary selection for robust networks that are resistant to change [19]. That said, the very fact that HMAs have therapeutic potential tells us that epigenetic changes play a strong supporting role in transformation. It is, therefore, imperative to understand the interplay between epigenetic and genetic lesions in the development of neoplasia. Here, we have been helped by an enormous body of recent work on a pre-neoplastic condition known as clonal hematopoiesis of indeterminate potential (CHIP).

## 4. Lessons from Clonal Hematopoiesis

The fact that around a third of all AML cases carry mutations in either the DNMT3A [21] or TET2 genes [22] implies a strong connection between methylation control and leukemogenesis. It was, therefore, surprising to find that many of us carry expanded clones of hematopoietic cells marked by these very mutations without ever developing overt neoplastic disease [23], a condition known as clonal hematopoieis of indeterminate potential (CHIP) [24]. CHIP clones are present at allele frequencies of 2% or more in approximately 15% of those aged 70 or over [23], while clones of lower allele frequency are so common that they can be regarded as a normal sign of aging [25,26,27]. While CHIP does carry an increased risk of progression to myeloid neoplasia, this only occurs at a rate of around 1% per year [24], telling us that mutations in genes encoding key components of the methylation machinery may facilitate the development of neoplastic disease, but are clearly not strongly transforming in their own right.

The high incidence of sub-clinical CHIP poses two key questions: First, how do lesions in epigenetic control mechanisms lead to persistent, low-level clonal expansion without causing overt transformation, and second, how can mutations in genes encoding such opposing activities as methylation (*DNMT3A*) and demethylation (*TET2*) have such similar consequences? CHIP is associated with an increased risk of atherosclerotic cardiovascular disease [23] and this has led to the identification of de-regulated inflammation as a common consequence of mutations in either *TET2* or *DNMT3A* [28]. Sterile inflammation also plays a role in myelodysplasia, where it is thought to contribute to disease progression [29]. It is, therefore, possible that the induction of an inflammatory environment contributes to the initial establishment of a CHIP clone. However, analyses of *TET2* and *DNMT3A* knockout mice point to a more intrinsic mechanism [30]. Mice carrying mull mutations in either gene have a proliferative advantage over their normal counterparts, resulting in accumulation of mutant stem cells in the bone marrow following competitive transplantation [31]. There are some interesting differences between the phenotypes, with null mutation of *DNMT3A* apparently increasing stem cell self-renewal without boosting progenitor cell production, while null mutation of *TET2* does both, at least in the short term [31]. A particularly interesting result of these experiments was that mice transplanted with DNMT3A mutant progenitor populations were later found to harbour DNMT3A mutant cells with a stem cell phenotype, indicating a degree of epigenetic plasticity, or the ability of DNMT3A mutated cells to revert from a progenitor back to a stem cell state [30,31]. This supports a decisive effect of the mutations on the intrinsic control of commitment versus self-renewal in hematopoietic stem cells.

Commitment and differentiation are presented above as a succession of changes in gene expression in which gene regulatory switches are thrown by the methylation/demethylation of regulatory sequences, the recruitment of transcription factors and the modification of histone proteins. It can be assumed that evolution has tuned the relevant switches to be sufficiently rapid, accurate and coordinated to ensure that each cell makes a clear-cut distinction between alternative fates. One way of doing this would be to combine competing activities into physical complexes that are regulated in a coordinated fashion, as has been reported for lineage specific transcription factors [32,33] and as is described below for a methylation complex [34]. Where coordinated systems such as these have evolved to function smoothly and quickly, inactivating mutations are likely to cause delays and indecision, regardless of whether they affect methylation (*DNMT3A*) or demethylation (*TET2*). This means that coordinated epigenetic switching can be regarded as a common function of *TET2* and *DNMT3A* that is likely to be affected detrimentally by mutations affecting either gene. A simple delay in commitment that occurs as a result of inefficient switching in this way might increase the output of a clone by permitting extra rounds of proliferation. However, a delay alone would be insufficient to explain the gradual increase of clone size over time that is typical of CHIP [35]. This steady increase tells us that the CHIP mutations must result in at least some additional self-renewal at the stem cell level [30]. In Figure 3, we use the image of a commitment step being slowed to a commitment slope to consider how mutations in the methylation machinery might increase the chance of self-renewal and pave the way for neoplastic transformation.

## 5. Neoplastic Transformation

The evolution from CHIP to neoplasia occurs as the result of further mutations in cells capable of self-renewal. CHIP mutations that affect epigenetic coordination of commitment are likely to prepare the ground for neoplastic progression firstly by increasing mutation rate (for instance by affecting the inflammatory environment and/or DNA damage response) and secondly by expanding the pool of cells that can undergo self-renewal. Furthermore, the “reflux” self-renewal loop sr2 (Figure 3C) is predicted to include a stage of relatively high activity that is absent from the normal loop sr1. This active state would effectively expand the repertoire of target genes for mutations that bestow a further selective advantage. In this way, the reduced stringency of gene regulation and expanded progenitor pool in a CHIP clone would be expected to increase the chances of further genetic/epigenetic changes that reinforce the reflux type self-renewal (sr2) and/or establish a new self-renewal loop at the progenitor level (sr3), for instance by blocking differentiation. Each of these events leads to a further selective advantage for the affected clone.

In this model, while mutations in epigenetic regulators such as *DNMT3A* or *TET2* can provide a basis for neoplasia, the truly transforming mutations that subsequently uncouple proliferation from differentiation target a wide range of other genes. Nonetheless, it is important to remember that all of these ultimately exert their transforming effects by interfering with the normal networks of hematopoietic gene regulation, effectively rewiring the epigenetic framework underlying Waddington’s landscape to create folds and pockets where aberrant self-renewal can take place. Epigenetic reprogramming is, therefore, a common property of all leukemias, and not just those that carry epigenetic mutations.

## 6. The AZA Shake-Up: An Interpretation of HMA Action

The fact that epigenetic rewiring is a common and evolving feature of all neoplastic transformation goes some way to explaining why treatment with hypomethylating agents can have effects across a wide spectrum of subtypes and stages of myelodysplasia and myeloid leukemia. The model presented in Figure 3 also suggests why the response to HMAs is largely independent of the presence of mutations in *TET2* or *DNMT3A* themselves [36], since these are early facilitators rather than late drivers of transformation. However, it does not explain why hypomethylation tends to be beneficial, rather than disruptive. Although experience tells us that this is the case it is not intuitive, since a reduction in methylation might be expected to result in a general deregulation and reshuffling of genetic networks. On the face of it, this is just as likely to make the situation worse (by activating proto-oncogenes) as it is to make it better (by reactivating tumour suppressor genes).

Here, it helps to remember that the normal valleys of commitment and differentiation in the epigenetic landscape of hematopoiesis are shaped by gene regulatory networks that have been selected over many millions of years of evolution to be robust to perturbations, or “canalised” [37]. In comparison, the non-conform regulatory networks that arise during neoplastic transformation are likely to be more precarious, with deviant regulators being poorly integrated [19]. Based on this, we suggest that the tendency of HMAs to be broadly beneficial rather than disruptive is a consequence of the deviant parts of a network being the most susceptible to epigenetic reshuffling. In a cell that has been trapped into an abnormal state of neoplastic self-renewal, this superficial reshuffling may allow the cells to reverse out of the trap and resume a more normal, canalised fate.

We illustrate this view in Figure 4, by considering the regulatory networks governing self-renewal and differentiation of a neoplastic stem cell expressing a potentially disruptive oncoprotein. Prior to commitment, stem cells are known to co-express genes relevant to each of their potential fates [9,10,11]. Initially, these programmes are not functionally engaged, but are held at the ready in a primed state [11]. As commitment takes place, one lineage programme crosses an activation threshold and is fixed in such a way that suppresses competing lineage programmes [38] while linking subsequent proliferation with progressive differentiation [39]. In the example shown, the stem cell expresses an oncoprotein with a certain probability of interfering with the chosen lineage programme. Even if this is a rare event, any cell affected may be forced to deviate from the normal path and enter a differentiation blocked, proliferative state of neoplasia in which selection favours the promoter methylation and consequent inactivation of tumour suppressor genes [40]. By effectively reversing this methylation, HMAs are thought to push the neoplastic cell back into a non-neoplastic state, leading to differentiation or death [41]. Figure 4 presents this classic view from a slightly different angle, based on the assumption that HMAs undo a number of superficial epigenetic connections all at the same time and thus reverse the neoplastic cell out of the epigenetic pocket in which it has become trapped, allowing the intrinsic programmes to then have another run at getting things right.

In this case, although the “aza shake-up” would give the cells a second chance, they nonetheless retain the oncoprotein, and therefore, their original tendency to make mistakes. Indeed, the fact that relapse remains a major problem following HMA therapy shows that a proportion of cells often retain (or regain) the leukemic phenotype, even through repeated cycles of treatment.

## 7. Strategies for Predicting and Improving Response

Despite high relapse rates, a proportion of patients clearly do experience real, long-term benefits from HMA therapy, even after other treatments have failed. This makes it vitally important to identify factors that influence response and outcome as well as ways of increasing the efficacy of treatment.

A number of studies have looked for an influence of mutations in specific genes on the response and outcome of MDS, CMML or AML patients treated with HMAs. Although associations have been reported with mutations in a variety of genes [42,43,44], most are weak and there is often inconsistency between studies [4]. Most importantly, there is no clear influence of DNMT3A mutations on HMA response, while TET2 mutations appear to be relevant only above 10% variable allele frequencies and when considered in conjunction with other markers [43,45]. The importance of the context, rather than just the presence of a particular mutation has been confirmed in an AI-based analysis of mutations over 29 loci in a group of 433 MDS patients [46]. By focussing on the 42% of patients who have 3 mutations or more, this study found around a third to have a combination predictive of a lack of response to HMA. A related strategy has been to look directly at the degree of CpG methylation itself, either at specific loci or dispersed throughout the genome on repetitive elements. As is the case for gene mutations, single loci do not provide reliable indicators and the best associations are obtained by integrating methylation states over multiple loci [47,48].

The response of myeloid neoplasia to HMA therapy, therefore, appears to be a complex phenotype influenced by multiple interacting features. Recent studies are extending beyond the intrinsic genetic and epigenetic signatures of the neoplastic clone itself to consider other parameters of potential relevance to outcome, such as metabolic markers, and interactions between the neoplastic clone and it’s environment [4]. For instance, it has recently been reported that AZA treatment of bone marrow MSCs both from healthy individuals and from patients with myelodysplasia improves their capacity to support hematopoietic cells in vitro [49] while AZA also reactivates the expression of endogenous proviruses, inducing an interferon response that can stimulate the immune reaction against neoplastic cells [50]. A better understanding of how extrinsic mechanisms such as these contribute to therapy success is likely to be important both for the prediction and the future improvement of response.

The fact that HMAs show activity across a range of clinical conditions in the absence of a single reliable molecular predictor of response is consistent with the Aza Shake-Up model suggested in Figure 4, in which HMAs help to reverse transformed cells out of whichever dead-end they find themselves in and give them another chance to resolve the networks concerned. It follows that the combination of HMAs with agents that improve either the correction or the elimination of cells reactivated in this way may have synergistic effects. The most promising HMA partners in this respect include the BCL2 inhibitor venetoclax [51], the PDL1 antibody nivolumab [52] and the CD47 antibody magrolimab [53]. In the case of BCL2 inhibition by venetoclax, one can imagine how shaking up the epigenetics while simultaneously lowering the barrier to apoptosis can increase the proportion of neoplastic cells that redirect down a death pathway following HMA treatment. Indeed, the efficacy of this combination is such that it has already become standard of care for elderly or unfit AML patients [51]. Both nivolumab and magrolimab follow a complementary approach by inhibiting immune escape signals that are expressed on HMA resistant cells, thus restoring their susceptibility to immune attack [54].

## 8. RNA Methylation

We have focussed up to this point on the effects of HMAs on DNA methylation. This focus is entirely justified in the case of DEC which, as described above, is processed intracellularly to 5-azadeoxyctidine triphosphate and incorporated into replicating DNA [55] where it captures, cross-links and inactivates the DNMT1 and DNMT3 enzymes responsible for CpG methylation of DNA. In the case of AZA, however, only around 15% of the drug that enters a cell eventually ends up in DNA, while the remaining 85% is incorporated into RNA [55]. Here it encounters and inhibits the enzyme DNMT2. DNMT2 is a relative of the DNA methyltransferases and uses essentially the same reaction mechanism but targets RNA rather than DNA. Accordingly, DNMT2 is inhibited by AZA in the same way that DNMT1 and DNMT3 are inhibited by DEC, resulting in significant cellular effects of AZA at the level of RNA methylation [56].

Cytosine C5 methylation is just one of a wide range of modifications that can be made to RNA [57]. Furthermore, there is an entire family of enzymes that, like DNMT2, direct cytosine C5 methylation [58], so that DNMT2 is responsible for only a fraction of the total 5meC in RNA. Nonetheless, the high conservation of DNMT2 across species argues for an important role in cell biology. Given this, it was initially surprising that mutation of the *dnmt2* gene had no obvious effect in a range of model organisms under laboratory conditions [59]. However, subsequent studies have suggested that Dnmt2 has important functions under stress conditions that are probably more relevant for evolutionary selection in the wild than for survival and reproduction in a controlled laboratory environment [60]. Eventually, a thorough re-examination of *dnmt2* knock-out mice backcrossed to homogeneity into a C57BL/6 background showed that, while adult mice are normal and fertile, there is a significant delay in osteo-hematopoietic development during the immediate post-natal period [61]. This is encouraging, since the requirement for Dnmt2 in an expanding postnatal osteohematopoietic system in mice would be consistent with specific effects of DNMT2 inhibition in human leukemia. Further research has now implicated both messenger RNA (mRNA) and transfer RNA (tRNA) targets of DNMT2 in the regulation of hematopoiesis [34,61,62].

## 9. DNMT2 Methylation of mRNA

The relevance of specific mRNA targets of DNMT2 has emerged following the independent identification of Dnmt2 among a set of RNA binding proteins that affect the lineage commitment of hematopoietic progenitors in mice [34]. By tracing a series of interactions involving both RNA- and DNA-binding proteins, Cheng et al., found that Dnmt2 methylates the freshly transcribed (nascent) 5′ end of certain mRNAs involved in hematopoietic lineage commitment. This methylation precipitates the formation of an mRNA/protein/DNA chromatin complex over the promoter region that then drives further transcription initiation. In this way, the Dnmt2-dependent complex sets up a positive feedback loop, fixing the active transcription of a key regulator once it has reached a set threshold and thus preventing the cell from going back on a decision once taken. Positive feedback loops of this sort clearly have the potential to resolve binary fate decisions quickly and cleanly by tipping them one way or the other and it is easy to see how they could help resolve lineage commitment decisions by mediating the rapid and decisive switching of gene expression as described above in the context of DNA methylation. Indeed, the Dnmt2-initated chromatin complexes described by Cheng et al. were also found to be associated with Dnmt1 and Tet2 proteins, suggesting possible co-regulation of chromatin accessibility (via DNA methylation) and transcription initiation (via RNA methylation). Practically, this suggests that the inhibition of DNA methylation by DEC and the inhibition of mRNA methylation by AZA might converge from two different directions to loosen the epigenetic resolve of leukemic cells and initiate the aza shake-up shown in Figure 4. To the best of our knowledge, DEC and AZA have yet to be tested in combination in the clinic, although each has been trialled extensively in combination with other drugs [63].

## 10. DNMT2 Methylation of tRNA

While Dnmt2-mediated methylation of mRNA clearly has a role to play in lineage commitment, the evolutionary roots of the enzyme go much deeper: From single celled organisms up to humans, Dnmt2 directs the C5 methylation of a small number of tRNAs (primarily tRNA-Asp) at position C38 in the anticodon loop, suggesting that DNMT2 plays an important role in the control of protein translation. In support of this, a recent shRNA screen for genes required to maintain cell survival in the presence of AZA found that down-regulation of the histone acetyl transferase CBP (CREB Binding Protein) had a substantial impact on protein translation and acted synergistically with AZA to reduce cell viability, but showed no synergy with DEC [62].

The effects of C38-methylation of tRNA-Asp include an increased rate of tRNA/amino acid charging by aspartyl tRNA synthetase [64], increased tRNA stability [60] and an improved fidelity of anticodon/codon interaction that serves to minimise mistranslation of glutamate codons to aspartate and *vice-versa* [61]. Given that the effects of all these changes on translation would be expected to be positive, it is interesting that methylation is not constitutive. This suggests that non-methylated tRNA-Asp has a role to play under certain conditions and that Dnmt2 mediates an adaptive response.

Consistent with this, early work on fission yeast (*Schizosaccharomyces pombe*) found that Dnmt2 (Pmt1) failed to methylate tRNA-Asp efficiently in standard growth media, but did so when the yeast was grown in medium containing peptone as a major carbon source [65]. A link between Dnmt2 activity and nutrient availability would be consistent with the retardation of post-natal osteohematopoietic development found in *dnmt2*^−/−^ mice noted above [61], since this is a period of extreme proliferative activity in the establishing marrow in which an appropriate response to nutrient supply may be decisive. A requirement for Dnmt2 under stress conditions has also been described in *Drosophila*, in which the Dnmt2-mediated protection of tRNA sub-fragments from degradation is required for subsequent Dicer activity and an appropriate siRNA response [66].

There is, therefore, accumulating evidence that the evolutionarily ancient, DNMT2-dependent C38 methylation of tRNA-Asp is a central component of a highly conserved translational response to stress. The three direct consequences of C38 methylation for the tRNA-Asp: increased stability, more efficient aspartate loading and a higher fidelity of codon recognition, would all serve to ensure that methylated tRNA takes over rapidly from non-methylated tRNA at the ribosome following DNMT2 activation. The effects that this then has on the spectrum of protein translation are not yet well understood. However, it has been reported that null mutation of *dnmt2* in mouse fibroblasts results in a strong reduction in the translation of proteins containing poly aspartate runs and that these proteins tend to be involved in transcription and gene expression [64]. This is consistent with a role for Dnmt2 in biasing translation of functional subsets of proteins in response to a change in conditions. Whether Dnmt2-dependent translation is influenced simply by the incidence of aspartate codons, or whether there may be a differential effect on the two alternative aspartate codons (GAU and GAC) is still unclear [67]. The GAU and GAC codons are indeed used differentially in genes involved in divergent cell processes such as proliferation and differentiation [68] and the Dnmt2-mediated methylation at C38 is dependent on modification of the “wobble” base 34 that can differentially affect GAU vs GUC codon recognition [69]. Even so, there is as yet no direct evidence that Dnmt2 activity can affect the relative translation efficiency of GAU vs GAC. Whatever the mechanism, there is an emerging consensus that tRNA modification by Dnmt2 exerts a level of translational control which enables the cell to cope with stress conditions. It remains to be seen whether or not the stress conditions concerned are relevant to the action of AZA in a neoplastic bone marrow environment.

## 11. Summary and Outlook

In summary, we have learned in recent years that the mechanisms of action of DEC and AZA are overlapping but distinct (Figure 5). DEC becomes incorporated exclusively into DNA, where it results in a general inhibition of DNA methylation and shakes up epigenetic patterns, releasing cells from the transformation trap and giving them another chance to enter a programme of differentiation or death. A fraction of AZA does the same thing, but most of it finishes up in RNA, where it inhibits just one of a whole range of RNA modification enzymes, albeit a highly conserved one that methylates tRNA to direct a basic translational response to stress conditions and methylates mRNA to coordinate hematopoietic lineage commitment.

It is to be hoped that the emerging knowledge concerning DNA and RNA modification and the mechanisms of HMAs will enhance our prospects for further improving the potential of HMA therapies. Having optimised dosing schedules, the current focus is on testing combinations with other agents. The extension of this approach to triplet or even higher order combinations to target neoplastic cells at multiple levels is likely to have an increasing impact on the treatment of AML and MDS, and to open up new opportunities for personalised therapy [70]. As we go down this route, a sound knowledge of the full range of HMA activities in the context of neoplastic marrow will be increasingly important to our efforts to identify and optimise the most effective synergies in any given situation.

## Figures and Tables

**Figure 1 cells-11-02589-f001:**
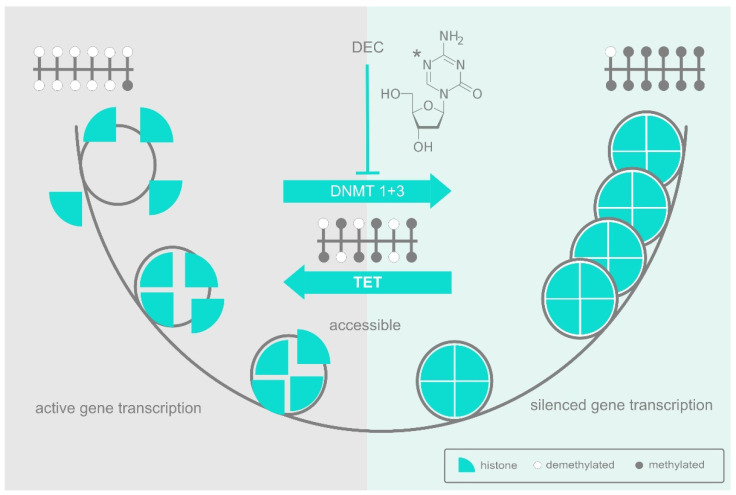
Azanucleosides affect DNA methylation and chromatin accessibility. A CpG island is shown at the top left of the figure in the predominantly non-methylated state (open circles) associated with active genes. Transcription factors can access their binding sites and modify histones to loosen histone/DNA interactions and open up the chromatin for transcription. In contrast, the CpG island at the top right is fully methylated (filled circles), preventing the binding of activating transcription factors. Tightly bound histones organise the DNA into nucleosomes and subsequently into higher order heterochromatin, locking in the inactive state. The CpG Island in the centre is in a dynamic, semi-methylated state subject to further methylation by DNMT enzymes or to demethylation by TET. Decitabine (DEC), shown top centre with the C5 position of the pyrimidine ring marked (*), depletes DNMT enzymes and tips the balance towards demethylation.

**Figure 2 cells-11-02589-f002:**
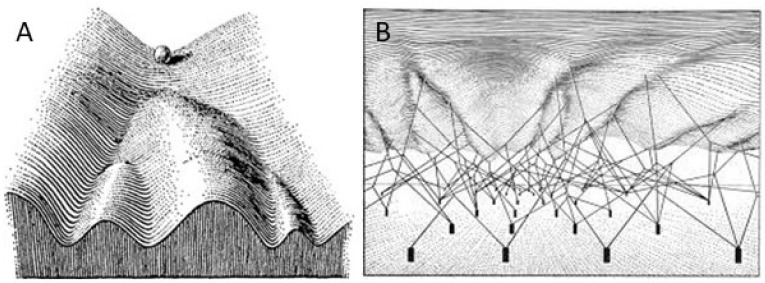
Waddington’s epigenetic landscape. Waddington’s epigenetic landscape (**A**) depicts the various potential routes via which a cell (Waddington: “part of the egg”) can develop as a system of valleys down which a ball can roll. (**B**) Looking from below, the landscape is defined by a framework of epigenetic “chemical interactions” emanating from fixed anchor points that represent the genes. Reprinted with permission from Ref. [15]. 1957, Taylor and Francis, London.

**Figure 3 cells-11-02589-f003:**
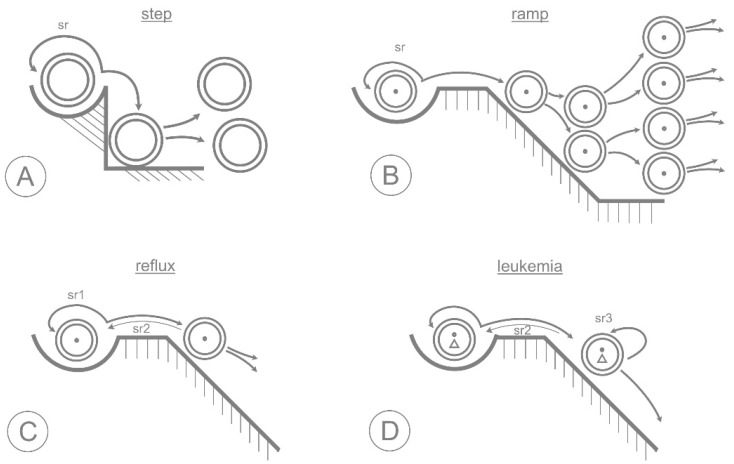
A model of CHIP and progression to neoplasia. (**A**) The Commitment Step: Normal stem cell commitment occurs as a rapid, irreversible step that requires coordinated rearrangement of DNA methylation patterns. Self-renewal in this case is limited to the niche (sr1). (**B**) The Commitment Ramp: A stem cell mutation (shown here as a dark circle) that slows epigenetic switching stretches the commitment step into a commitment ramp. Any cell division occurring during the prolonged commitment period would increase the clonal output. However, in this scenario there would be no increase of true self-renewal, and therefore, no continuous increase of clone size. (**C**) Reflux: Retardation of the commitment process increases the probability that a committing daughter cell reverts to a stem cell state. This is equivalent to symmetric self-renewal of mutated stem cells that, however rare, would steadily increase their representation in the stem cell pool. The inclusion of a committing progenitor stage in the new self-renewal loop (sr2) allows selection for further mutations that provide a selective advantage to either stem or committing cells. (**D**) Leukemia: As the clone becomes progressively more deviant, further mutations that block differentiation (open triangle) allow self-renewal of progenitors without recourse to the stem cell state.

**Figure 4 cells-11-02589-f004:**
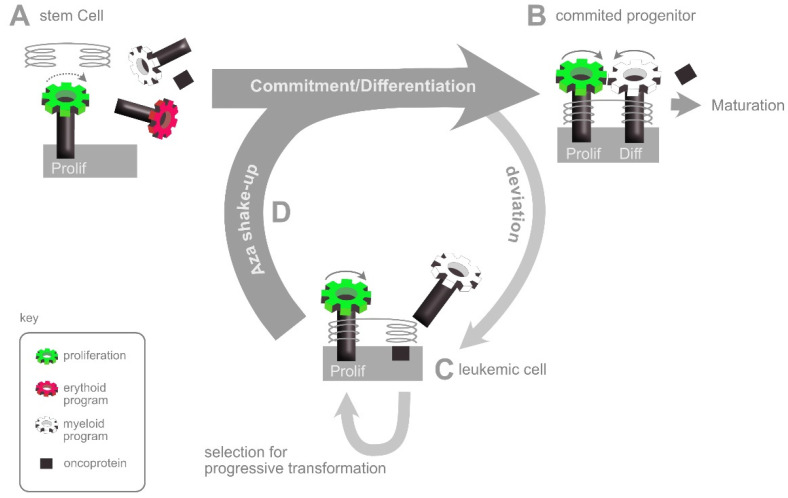
The aza-shake-up. (**A**) A multipotent stem cell maintains the genetic programmes relevant to each potential fate in a primed state in which the chromatin is accessible but gene expression low. Proliferation is slow (dashed arrow) and is not linked to differentiation. (**B**) During lineage commitment, one lineage programme is activated and others silenced. Proliferation is linked to differentiation to limit expansion. (**C**) An oncoprotein interferes with the differentiation programme to disengage proliferation from differentiation. Even if this occurs in only a small proportion of the mutant cells, these can become trapped in a self-renewal loop that provides the basis for amplification and further selection. (**D**) Azanucleoside treatment reactivates (recently) suppressed genes, resulting in an epigenetic shake-up and the chance to settle back into a normal fate.

**Figure 5 cells-11-02589-f005:**
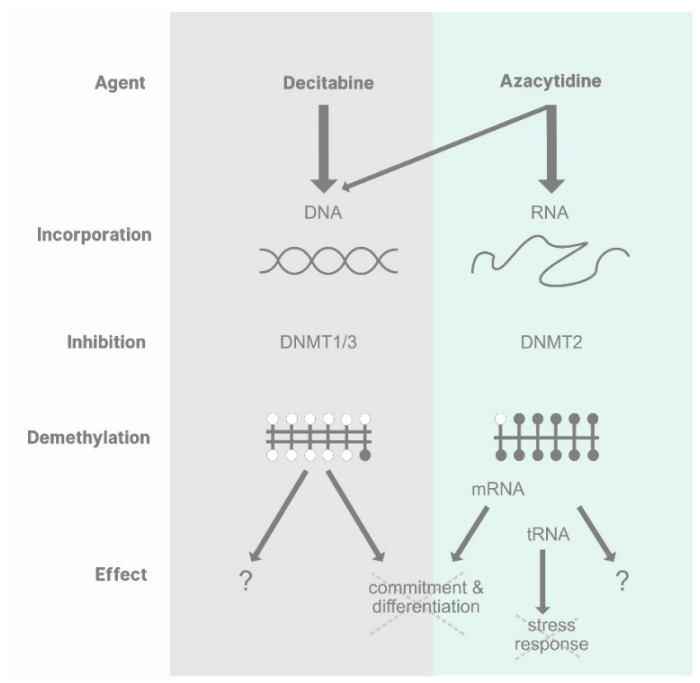
Decitabine: Intracellular processing into 5-aza-2′-deoxycytidine triphosphate and incorporation into DNA results in inhibition of DNMTs 1 and 3. This leads to an overall decrease of DNA methylation and the reactivation of superficially silenced genes, allowing a proportion of neoplastic cells to exit their aberrant epigenetic state (see Figure 4). Azacytidine: a fraction of AZA is converted intracellularly into 5-aza-2′-deoxycytidine triphosphate and follows the same fate as DEC. However, the majority becomes incorporated as 5-azacytidine into RNA, where it inhibits DNMT2. This leads to a specific decrease of RNA methylation at DNMT2 sites only. The specific hypomethylation of DNMT2 sites in mRNAs can interrupt positive feedback loops of transcription and reverse commitment decisions. The specific hypomethylation of DNMT2 sites in some tRNAs may make the cell more susceptible to stress by compromising their ability to raise a translational stress response.

## Data Availability

Not applicable.

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
