# Peer review of "How Azanucleosides Affect Myeloid Cell Fate"

_cells, 2022, doi:10.3390/cells11162589_

Round 1
Reviewer 1 Report
In this review, the authors provide a comprehensive and informative appraisal of the molecular mechanisms and mode of action of decitabine versus azacytidine. The authors clarify well the differences between these two azanucleosides and the potential implications for therapy of myeloid neoplasms. I have only a few comments.
MAJOR ISSUES
1. A paragraph on the search for molecular predictors of decitabine response versus azacytidine response would be of interest. Although consolidated molecular predictors may not yet be available for this purpose, a few sentences informing the reader on which research efforts are in progress would be important.
2. In the “Summary and Outlook” section, the authors should mention the biological and clinical rationale for combining azanucleosides with innovative monoclonal antibodies for myeloid neoplasia, referring to a very recent review on this important topic (Gallazzi et al. New Frontiers in Monoclonal Antibodies for the Targeted Therapy of Acute Myeloid Leukemia and Myelodysplastic Syndromes. Int. J. Mol. Sci. 2022, 23, 7542. https://doi.org/10.3390/ijms23147542). Many clinical trials of combination therapy exploiting azanucleosides and innovative monoclonal antibodies have been conducted and/or are in progress, and this represents an important perspective that deserves to be mentioned.
MINOR ISSUES
1. In the legend to Figure 5, the word “Summary” may be deleted.
Author Response
We thank the reviewer for their overall positive appraisal of our work and the suggestions made.
Point 1: A paragraph on the search for molecular predictors of decitabine response versus azacytidine response would be of interest. Although consolidated molecular predictors may not yet be available for this purpose, a few sentences informing the reader on which research efforts are in progress would be important.
Response 1: In the original version of our manuscript, we had referred very briefly in the final paragraph of section 6 to intrinsic and extrinsic factors influencing response. We have now expanded this to cover the issue of molecular predictors of response and included it as a new section 7 following on from the Aza-Shake-Up section.
Point 2: In the “Summary and Outlook” section, the authors should mention the biological and clinical rationale for combining azanucleosides with innovative monoclonal antibodies for myeloid neoplasia, referring to a very recent review on this important topic (Gallazzi et al. New Frontiers in Monoclonal Antibodies for the Targeted Therapy of Acute Myeloid Leukemia and Myelodysplastic Syndromes. Int. J. Mol. Sci. 2022, 23, 7542. https://doi.org/10.3390/ijms23147542). Many clinical trials of combination therapy exploiting azanucleosides and innovative monoclonal antibodies have been conducted and/or are in progress, and this represents an important perspective that deserves to be mentioned.
Response 2: We consider the new section 7 to be a good place for a somewhat more detailed coverage of the combination of HMAs with other agents and in particular with innovative antibodies, as suggested by the reviewer. To avoid repetition, we have also removed the brief mention of combination therapies from the Summary and Outlook section, and rewritten this accordingly.
Point 3: In the legend to Figure 5, the word “Summary” may be deleted
Response 3: This has now been done.
Reviewer 2 Report
The introduction is well written and interesting. The introduction gives the expectation that in the review the biological activity of DEC and AZA is revealed in the context of the observed response depth and duration in patients, and which drug combinations and dosing schedules are most likely to give benefit for the patient in relation to the biological activity of DEC and AZA.
The part on “DNA methylation and DNA methyltransferases” is nicely written and explains very well the process of DNA methylation and regulation of gene transcription. The last sentence of this part “the more advanced the methylation throughout a region, the less likely it is to be reactivated” needs a reference. It is unclear how advanced methylation is defined, and if this is just made up by the authors or that there is scientific proof for this. In Figure 1; I would change the term lollypops into open and closed circles, and change the word “locked”, as opposed to active, into silenced gene transcription.
The part “The epigenetic landscape” is wordy and long-winded, and not very well connected to the treatment of AZA and DEC. Try to make it more direct, leaving out words/sentences not needed for this review.
The part “Lessons from CHIP” needs more references as it now seems all like speculations. There are a lot of publications on CHIP. For example, that CHIP cells, normal cells with a TET or DNMT3A mutation, have a proliferative advantage should be first introduced, with the references included. The statements and speculations are not deduced or connected to published data on clonal hematopoiesis, lineage commitment and proliferation and not to the many publications on HMA treatment in AML.
Also the part “The AZA shake up” needs references and more connection to the literature (till sentence 340). Now it is not a review. It is good to speculate, after discussing the results found till thus far, but his is over the top. Moreover, this part is wordy.
The introduction and abstract suggests that the review will deal with situations in where we should use DEC and in where AZA, and in combination with what, in relation to their mode of action, however, there is only a very small part directed to this in the review.
There is too much detail on DNMT2A is the review.
Author Response
Point 1: The introduction is well written and interesting. The introduction gives the expectation that in the review the biological activity of DEC and AZA is revealed in the context of the observed response depth and duration in patients, and which drug combinations and dosing schedules are most likely to give benefit for the patient in relation to the biological activity of DEC and AZA.
Response 1: We are glad that the reviewer found the introduction interesting. The reviewer’s expectation of a clinical analysis highlights a misunderstanding, for which we must take full responsibility. We had never intended to extend our coverage to include specific treatment recommendations. Our aim was to complement rather than repeat recent reviews that approach similar issues from a more clinical perspective. Our priority was to look at the ways in which HMAs act at the level of cell biology and thus to provide a perspective against which clinical effects can be interpreted. We apologize for having given the wrong impression. We have now critically appraised the abstract and introduction and found two parts that we believe led to this misunderstanding. These have now been modified in order to make our intentions absolutely clear.
Abstract:
Original: We also consider how azanucleosides can act to reset epigenetic programmes in a wide variety of neoplastic conditions, regardless of DNMT3A or TET2 mutations. We can see that this sentence might be taken to mean that we were intending to consider azanucleoside effects on various neoplastic conditions singly. We have now replaced this sentence with a more explicit statement: Revised: We also consider why the efficacy of azanucleoside treatment is not limited to neoplasias carrying mutations in epigenetic regulators.
Introduction:
Original: Given the widespread use of the azanucleosides and the growing relevance of combination therapies, it is now becoming increasingly important to understand precisely how the HMAs affect cell biology. This information will be key to determining which variables influence the probability, depth and duration of a clinical response and which drug combinations and dosing schedules are most likely to provide a benefit in any given situation. With this in mind, we consider here our current understanding of HMA function from a cell biology perspective. Our intention here was to prepare the reader for a cell biology perspective that should help with evaluating the future results of trials that are becoming increasingly diverse and not to introduce a comprehensive, cell biology-based evaluation of the clinical experience to date. Here too, we have amended the passage to make our intention clearer:
Revised: As HMA-based treatments become more common and more diverse, it is going to be increasingly important to understand precisely how the HMAs affect cell biology. This information will be key to interpreting the results of current and future trials and determining which variables influence the probability, depth and duration of a clinical response in any given situation. Here, we aim to provide relevant background information by considering our current understanding of HMA function from a cell biology perspective.”
Point 2: The part on “DNA methylation and DNA methyltransferases” is nicely written and explains very well the process of DNA methylation and regulation of gene transcription. The last sentence of this part “the more advanced the methylation throughout a region, the less likely it is to be reactivated” needs a reference. It is unclear how advanced methylation is defined, and if this is just made up by the authors or that there is scientific proof for this.
Response 2: The referee has a very good point and on reflection we have to agree. We were guilty of oversimplifying a complex issue in an attempt to convey the concept that there are various levels of silencing and that deeply silenced genes are more difficult to reactivate. We were wrong to imply that this is simply an issue of DNA methylation. Rather than to qualifying the statement by going into the details of intragenic vs extragenic methylation, interdependencies between DNA methylation and histone modification, levels of chromatin packaging and the issue of reprogramming, we have simply deleted the sentence. We hope that this is an acceptable solution.
Point 3: In Figure 1, I would change the term lollypops into open and closed circles, and change the word “locked”, as opposed to active, into silenced gene transcription.
Response 3: We have followed the reviewer’s advice and made these changes as suggested
Point 4: The part “The epigenetic landscape” is wordy and long-winded, and not very well connected to the treatment of AZA and DEC. Try to make it more direct, leaving out words/sentences not needed for this review. Indeed. More clinical considerations of the link aza functions to specific treatment decisions
Response 4: As explained above, our main intention was to provide background information at the level of cell biology, rather than to summarise the experience with AZA and DEC in the clinic and this is now stated more clearly in the opening sections. We consider the section “The Epigenetic Landscape” to serve an important purpose in this respect. It is a central part of the review and it brings together a number of concepts that are far less widely appreciated than the landscape itself (the regulatory framework underlying the landscape; distortion to generate folds; the relative contribution of genetic and epigenetic changes etc.). We can see that this section might seem wordy and long-winded to an expert in the field who is already familiar with the concepts, but we believe that it supplies important background for those who might not yet be.
Point 5: The part “Lessons from CHIP” needs more references as it now seems all like speculations. There are a lot of publications on CHIP. For example, that CHIP cells, normal cells with a TET or DNMT3A mutation, have a proliferative advantage should be first introduced, with the references included. The statements and speculations are not deduced or connected to published data on clonal hematopoiesis, lineage commitment and proliferation and not to the many publications on HMA treatment in AML.
Response 5: We are very appreciative of this criticism. We have now rewritten this section largely as recommended by the reviewer, focussed some of the arguments and qualified statements with specific references. However, for the reasons outlined above, we have kept the focus on biology and not extended our coverage to include HMA treatment of AML.
Point 6: Also the part “The AZA shake up” needs references and more connection to the literature (till sentence 340). Now it is not a review. It is good to speculate, after discussing the results found till thus far, but his is over the top. Moreover, this part is wordy.
Response 6: We have now rewritten the “AZA shake-up” section to provide specific qualifying references where appropriate. We have also tried to distinguish more clearly between established knowledge and speculation/interpretation.
Point 7: The introduction and abstract suggests that the review will deal with situations in where we should use DEC and in where AZA, and in combination with what, in relation to their mode of action, however, there is only a very small part directed to this in the review.
Response 7: See our response to the first comment. This comment stems from a miscommunication / misunderstanding of our intentions, that we have now clarified in the earlier sections.
Point 8: There is too much detail on DNMT2A is the review.
Response 8: There is substantial evidence that DNMT2 is the major target of AZA. The role of DNMT2 in mRNA and tRNA methylation and the consequences for the lineage commitment and stress response are well described in model organisms but, in our experience, still only poorly appreciated by those who focus on HMA actions in the setting of human cancer. These activities are highly conserved throughout evolution and the indications are that they will turn out to be clinically highly relevant. Given that we aim to review the effects of HMAs on the level of cell biology (and have now made this more clear) we consider this degree of coverage to be justified and would prefer not to condense this part of the review.
In summary, we are truly appreciative of the considered and open criticism of the original version of our review and are very grateful to have has the opportunity to make it substantially clearer and more accessible to the intended readership. We hope that the revised version is now considered acceptable for publication.
Round 2
Reviewer 1 Report
The authors have addressed all the issues that had been raised. The manuscript has significantly improved. No further comments from my side.